# BCG Vaccine—The Road Not Taken

**DOI:** 10.3390/microorganisms10101919

**Published:** 2022-09-27

**Authors:** Coad Thomas Dow, Laith Kidess

**Affiliations:** 1Department of Ophthalmology and Visual Sciences, McPherson Eye Research Institute, Madison, WI 53705, USA; 2Mindful Diagnostics and Therapeutics, Eau Claire, WI 54701, USA; 3Department of Biochemistry, University of St. Thomas, St. Paul, MN 55105, USA

**Keywords:** *Bacillus Calmette-Guérin* (BCG), tuberculosis, non-tuberculous mycobacteria (NTM), nonspecific effects, trained immunity, type 1 diabetes, multiple sclerosis, Parkinson’s disease, Alzheimer’s disease, *Mycobacterium avium* ss. *paratuberculosis* (MAP), molecular mimicry, global burden of disease

## Abstract

The *Bacillus Calmette-Guérin* (BCG) vaccine has been used for over one hundred years to protect against the most lethal infectious agent in human history, tuberculosis. Over four billion BCG doses have been given and, worldwide, most newborns receive BCG. A few countries, including the United States, did not adopt the WHO recommendation for routine use of BCG. Moreover, within the past several decades, most of Western Europe and Australia, having originally employed routine BCG, have discontinued its use. This review article articulates the impacts of those decisions. The suggested consequences include increased tuberculosis, increased infections caused by non-tuberculous mycobacteria (NTM), increased autoimmune disease (autoimmune diabetes and multiple sclerosis) and increased neurodegenerative disease (Parkinson’s disease and Alzheimer’s disease). This review also offers an emerged zoonotic pathogen, *Mycobacterium*
*avium* ss. *paratuberculosis* (MAP), as a mostly unrecognized NTM that may have a causal role in some, if not all, of these diseases. Current clinical trials with BCG for varied infectious, autoimmune and neurodegenerative diseases have brought this century-old vaccine to the fore due to its presumed immuno-modulating capacity. With its historic success and strong safety profile, the new and novel applications for BCG may lead to its universal use–putting the Western World back onto the road not taken.

## 1. Introduction

Humans have had a close relationship with the bacterium that causes tuberculosis, *M. tuberculosis* (Mtb) for millennia [1]; discovered by Robert Koch in 1882, Mtb is responsible for more deaths than any other human pathogen [2,3].

BCG is the only vaccine currently available against tuberculosis (TB) with over four billion doses, it has been the most widely administered vaccine; in 2020, global BCG immunization coverage among 1-year-olds was an estimated 85% [4]. The efficacy of BCG is variable; and although it prevents infants from infection with severe forms of disseminated TB, it does not protect against the most common form, pulmonary TB [5]. To boost its protective response, several alternative vaccine candidates, including recombinant live vaccines for BCG replacement as well as subunit vaccines (viral vectored or based on adjuvanted recombinant proteins) are under development [6].

The first BCG vaccination was given in 1921 to an infant with an extreme risk of developing disseminated TB; his recovery spurred further use of BCG [7]. A significant setback occurred less than ten years later when BCG, contaminated with live *M. tuberculosis*, was given to 251 newborns in Lubeck, Germany [8]. Of the 251 inadvertently infected neonates, 173 developed TB and 72 died. The erroneous concern that BCG had reverted to a pathogenic bacterium and was responsible for the “Lubeck disaster” led to early vaccination skepticism about BCG.

Amplifying this skepticism was a variable response to the different strains of the BCG vaccine. When evaluating BCG, the United States Public Health Service chose the Tice strain of BCG vaccine; other countries such as the United Kingdom used the Copenhagen strain. While the Tice strain showed little benefit in the US trials the Copenhagen strain of BCG was found to be particularly effective against TB [9].

There were two hypotheses to explain this disparity: there were actual differences in BCG strains as differing strains have differing properties [10]; and/or different exposures in the states where the US Public Health Service conducted their trials: Alabama, Puerto Rico and Georgia. These populations have exposure to environmental mycobacteria. That exposure, as of the “hygiene hypothesis,” could have provided protection against TB that could not be improved upon by BCG [11]. This hypothesis suggests that input from microbes assists in setting up regulation of the immune system. The microbes collectively are our Old Friends and population from high-income countries with “high hygiene” are deprived from interaction with these Old Friends. Mycobacteria are a component of the Old Friends and absent or lessened early life exposure to mycobacteria may be contribute to the diseases addressed in this paper [12]. The influence of BCG as an Old Friend extendes to allergic disease as well; introduction of BCG for its immunomodulatory effect is felt to benefit allergy prevention as well as treatment [13,14].

Meanwhile, public policy in most of the rest of the world recommended routine vaccination BCG vaccination. Notwithstanding its difficult start, currently BCG vaccination is given to 140 million infants each year [15]. Noteworthy is a trial detailing the non-waning efficacy of BCG imparting lasting protection against tuberculosis for 50 to 60 years [16]. Moreover, in addition to BCG as part of the standard-of-care for non-invasive bladder cancer (discussed later), in a trial with a 60-year follow-up, BCG vaccination was associated with a significantly lower rate of lung cancer compared to placebo recipients [17]. Furthermore, direct intralesion BCG injection into metastatic melanoma lesions can be highly effective for the injected lesion as well as noninjected distal lesions [18]. BCG is derived from *M. bovis*, a closely related organism and part of what is termed the Mtb complex. *M. bovis* is mostly studied in ruminant animals yet also causes tuberculosis in humans [19]. BCG can also be protective for animals lessening disease in the animal and reducing the zoonotic risk [20,21].

Health consequences for those populations not receiving routine BCG are featured in this article; these include increases in tuberculosis, diseases caused by non-tuberculous infections, autoimmune disease and neurodegenerative disease (Figure 1).

## 2. Tuberculosis after BCG Discontinuation

Individual countries continuously monitor the efficacy of their BCG program; an international body that gives guidance regarding continuing/discontinuing routine BCG usage is the International Union Against Tuberculosis and Lung Disease (IUATLD). This policy agency has three indicators it recommends to decide discontinuation of universal BCG vaccination: average annual notification rate of sputum smear-positive pulmonary TB of ≦5 per 100,000 population over the previous 3 years; average annual notification rate of TB meningitis in children under 5 years old of <1 per 10 million general population over the previous 5 years; and average annual risk of tuberculosis infection of ≦0.1% [22]. Further identification of targeted groups comes via extension of this list compiled by the WHO, which aggregates high burden countries into three interrelated groups: high burden of TB, high HIV-associated TB burden countries and high multidrug/rifampin-resistant TB burden countries [23]. With decreasing TB over the past 40 years, several countries with “low burden” have ceased universal BCG vaccination [24]. Reporting of TB in these low-burden countries has been confounded by the fact that many of the cases were in patients from other countries-this a consequence of changing immigration policies [25].

## 3. BCG and Non-Tuberculous Mycobacteria

BCG is an attenuated live vaccine, and thus shares epitopes with mycobacteria other than tuberculosis—non-tuberculous mycobacteria (NTM); this provides a mechanism for cross-protection against infections from NTM [26]. NTMs are ubiquitous and can cause disease in susceptible individuals; there has been an increase in NTM disease in developed countries where routine BCG vaccination has been discontinued [27,28,29,30]. Cervical lymphadenitis is causally attributed to *M. avium intracellulare* complex (MAC). An NTM disease, cervical lymphadenitis has significantly increased since stopping BCG vaccination in France [31], the Czech Republic [32], Sweden [33] and Finland [34].

Leprosy is also caused by an NTM: *Mycobacterium leprae*. Although it is mostly viewed in an historic context, more than 200,000 new cases were recorded by the WHO in 2018 [35,36]. Protection provided by BCG vaccination against *M. leprae* is well recognized [37] as BCG decreases the risk of leprosy by 50% to 80% with the benefit improving with the number of BCG booster doses [38,39]. Another NTM disease, Buruli’s ulcer, caused by *Mycobacterium ulcerans*; it is a necrotizing skin disease. Buruli’s ulcer is the third most prevalent mycobacterial infection after tuberculosis and leprosy [40]. Buruli’s ulcer was described in 1948 in Australian patients [41]; this NTM disease is found primarily in poor areas of Africa; the Congo [42] and Uganda [43] and increasingly so in West Africa [44,45,46]. BCG vaccination of infants protects the recipients as children and later as adults from the serious osteomyelitis that is associated with Buruli’s ulcer [47]. Protection against Buruli’s ulcer provided by of BCG, as shown by prospective trials, is significant with overall protection rate protection of 47% [48,49].

A temptingly parsimonious plausibility is that the BCG-benefited diseases featured in this article are the indirect result of infection by another NTM that shares antigens with BCG [50,51]. Associated with autoimmune diabetes, multiple sclerosis and Parkinson’s disease is *Mycobacterium avium* ss. *paratuberculosis* (MAP) [52,53].

MAP has been proposed to have a causal role in Alzheimer’s disease as well [54]. Supporting an infectious contribution to Alzheimer’s is the recognition that the Alzheimer’s-associated amyloid beta protein is an antimicrobial peptide [55]. Moreover, the pathognomonic Alzheimer’s plaque constructed by microglia parallels macrophage construction of granulomas and is evocative of mycobacterial granulomas [56].

MAP is thought to have gone through an “evolutionary bottleneck” along with *M. tuberculosis* and *M. leprae*, the agents that cause TB and leprosy [57].

## 4. BCG and Autoimmune Disease

BCG has newfound therapeutic potential in the common autoimmune diseases type 1 diabetes (T1D) and multiple sclerosis (MS). BCG lowers blood sugar in diabetics and delays disease progression in MS possibly via immune stimulation [58]. A world map demonstrating countries with routine BCG usage is virtually the inverse of the world map of autoimmune disease prevalence (Figure 2).

### 4.1. BCG and Type One Diabetes (T1D)

T1D is mostly a disease of childhood and young adults and occurs with immune-mediated destruction of the insulin-producing cells of the pancreas [60]. In 2018, Dr. Denise Faustman presented favorable data regarding BCG vaccination in T1D patients at the American Diabetes Association (ADA) Scientific Sessions; her study subsequently was published in the medical journal *npj Vaccines* [61]. The Harvard scientist reported a follow-up study of participants with long-standing type 1 diabetes (T1D) that were treated with the BCG vaccine. This repurposed use of BCG was an extension of her previous work using BCG in an animal model of T1D [62].

BCG restored blood sugars to near normal; remarkably, this was seen even in patients with advanced disease of greater than twenty years duration. Mechanistically, this effect was proposed to have been driven by a reset of the immune system accompanied by a shift in glucose metabolism; this shift is from oxidative phosphorylation in which there is minimal sugar utilization for energy production to aerobic glycolysis in which there is high glucose utilization for energy production [63].

### 4.2. BCG and Multiple Sclerosis (MS)

MS is a central nervous system (CNS), immune-mediated, inflammatory disease characterized by demyelination [64]. Worldwide, MS affects more than 2.8 million individuals and it most often afflicts young adults [65]. Pathologically, T and B lymphocytes that are activated in the periphery migrate to the CNS where they produce demyelination and local inflammation [66]. Animal studies testing BCG against the pathology of MS have used the valuable experimental autoimmune encephalomyelitis (EAE) model of MS [67].

Though the cause of MS is unknown, studies have shown that BCG vaccination imparts beneficial reduction in MS disease activity by modulating T cell-mediated immunity [68]. In clinical trials, administration of a single dose of BCG reduced the magnetic resonance imaging (MRI) activity in relapsing–remitting MS patients, a common form of MS [69]. Clinically isolated syndrome (CIS) is an initial presentation of characteristic inflammatory demyelination that has not progressed to fulfill the diagnosis of MS; in CIS, when studied over a 5-year period, BCG vaccination delayed the second demyelinating episode [70].

## 5. BCG and Neurodegenerative Disease

Accumulating data suggest a critical role played by the immune system in neurodegenerative Alzheimer’s and Parkinson’s diseases [71]. Exposure to BCG in elderly adults showed 58% reduced risk of developing AD and 28% reduced risk of developing PD [72].

### 5.1. BCG and Alzheimer’s Disease (AD)

Worldwide, the most common cause of dementia is AD [73]. AD is characterized by abnormal protein deposits: the extracellular cerebral deposition of β-amyloid (Aβ) peptides and intracellular neurofibrillary tangles of tau with a juxtaposition of cerebral inflammation [73].

In a recent population study an inverse relationship was found between the incidence of Alzheimer’s disease and vaccination with BCG. In countries with high BCG usage, even after adjusting for factors such as longevity and wealth, there was a lower prevalence of AD. A beneficial modulation of the immune system imparted by BCG was the authors’ hypothesis resulting in a decreased prevalence of AD [74]. This is supported by animal studies where BCG vaccination was associated with an increase in anti-inflammatory CNS response resulting in an improvement in cognitive function [75].

Intravesicular BCG is part of the standard-of-care for bladder cancer in which the cancer has not invaded the bladder muscle [76]. The course of bladder cancer patients who received BCG were compared to bladder cancer patients for whom BCG was not part of their recommended treatment [77]. The results showed bladder cancer patients treated with BCG were significantly less likely to develop AD compared to those not similarly treated. The bladder cancer mean age was 68 years and the mean age for AD diagnosis was 18 years later, at 84 years. A dramatic reduction was seen in AD risk was seen in those receiving BCG: BCG treatment imparted four-fold less risk for developing AD compared to those not treated with BCG. The authors suggested that confirmation of their retrospective population study would support prospective studies of BCG in AD [77]. In a follow-up, multi-cohort study again it was shown that intravesicular BCG imparted a protective benefit against risk of AD; interestingly, it also showed protection against Parkinson’s disease [72]. A recent open-label, non-placebo-controlled study employing BCG in cognitively normal participants showed a reduction of AD risk as measured by plasma amyloid [78].

These studies mirror investigations that show benefit of a variety of vaccinations in AD [78,79,80,81,82].

### 5.2. BCG and Parkinson’s Disease (PD)

PD is the second most common neurodegenerative disease [83]. The neuropathology of PD is characterized by a loss of specific pigmented dopaminergic neurons in regions of the brain associated with PD, the substantia nigra pars compacta (SNc); this is accompanied by an abnormal accumulation of α-synuclein protein called Lewy bodies—a form of intraneuronal inclusions present in PD and another neurodegenerative disease, Lewy body dementia [84]. Accumulating evidence supports a role for the inflammatory response in PD pathogenesis characterized by highly active microglia [85].

A murine model of PD, 1-methyl-4-phenyl-1,2,3,6-tetrahydropyridine (MPTP) treated mice, has become a widely used and valuable model for PD investigations [86]. In the MPTP model, BCG vaccination induces Treg responses that suppress inflammation and preserve the striatal dopaminergic system in BCG-treated mice; in doing so, BCG offers neuroprotection in this animal model of PD [87].

While there are no current trials employing BCG in PD [88], the promising population study showing a significant reduced risk of PD after BCG [72] suggests the value of such a trial. Novel blood-based PD biomarkers will likely aid in the assessment of interventional use of BCG [89].

## 6. Discussion

The BCG vaccine was developed over one hundred years ago; it is the most employed vaccine and has not undergone modifications. Notwithstanding, the BCG vaccine has protected many millions from the severe, disseminated forms of TB. Moreover, via cross-mycobacterial effects against non-tuberculous mycobacteria, BCG has protected against diseases caused by NTM. Increasingly recognized is BCG’s off-target effects against other infections and diseases; these are referred to as non-specific or heterologous effects. This was seen in the early use of BCG; in 1931, Calmette reported a 4-fold reduction in deaths due to nontuberculous infection during the first year of life in children immunized with BCG [7]. In the decades that followed, a reduction in all-cause childhood mortality associated with BCG was found in several studies [90]. This beneficial effect was also found in the elderly who were hospitalized for infection; BCG protected the elderly who received BCG or placebo from new infections in at the time of discharge [91].

In 2014, the state of BCG expanded therapeutic use was articulated by Netea:

“… despite the epidemiological evidence for heterologous protective effects of BCG vaccination, the perceived lack of biological plausibility has been a major obstacle in recognizing and in investigating these effects.”[92]

What then is the biological plausibility as to how BCG exerts its benefits on this wide array of seemingly unrelated diseases? In addition to immunologic memory induced via the adaptive immune response, BCG imparts heterologous protection via the innate immune response. This includes vaccine-induced immune and metabolic alterations and epigenetic reprogramming of innate white cell populations resulting in heightened responses to subsequent stimuli, this has been named “trained immunity” [93]. The specific change in cellular metabolism associated with immune activation involves a shift from oxidative phosphorylation to glycolysis [94].

The global burden of disease (GBD) is a comprehensive effort to assess worldwide disease epidemiologic levels and trends the results of which are meant to inform policymakers. Below are the results from a cursory internet search for the GBD for each of the primary diseases featured in this article:

### 6.1. T1D

There is good evidence that the incidence of type 1 diabetes among children is increasing in many parts of the world. The International Diabetes Federation’s *Diabetes Atlas*, 5th edition, estimates that increase to be 3% per year [95].

### 6.2. MS

“Multiple sclerosis is not common but is a potentially severe cause of neurological disability throughout adult life. Prevalence has increased substantially in many regions since 1990.”[96]

### 6.3. AD

“We estimated that the number of people with dementia would increase from 57.4 million cases globally in 2019 to 152.8 million cases in 2050.”[97]

### 6.4. PD

“Over the past generation, the global burden of Parkinson’s disease has more than doubled as a result of increasing numbers of older people, with potential contributions from longer disease duration and environmental factors. Demographic and potentially other factors are poised to increase the future burden of Parkinson’s disease substantially.”[98]

These sobering assessments coupled with the knowledge that BCG, with its newfound broad utility and a hundred-year safety history, should prompt large-scale clinical trials attempting to bend the curve away from the projected global disease burden from T1D, MS, PD and AD. Regardless of genetic and/or environmental contributions to these diseases, BCG vaccination may be just the thing to pave the road to improved global health.

## Figures and Tables

**Figure 1 microorganisms-10-01919-f001:**
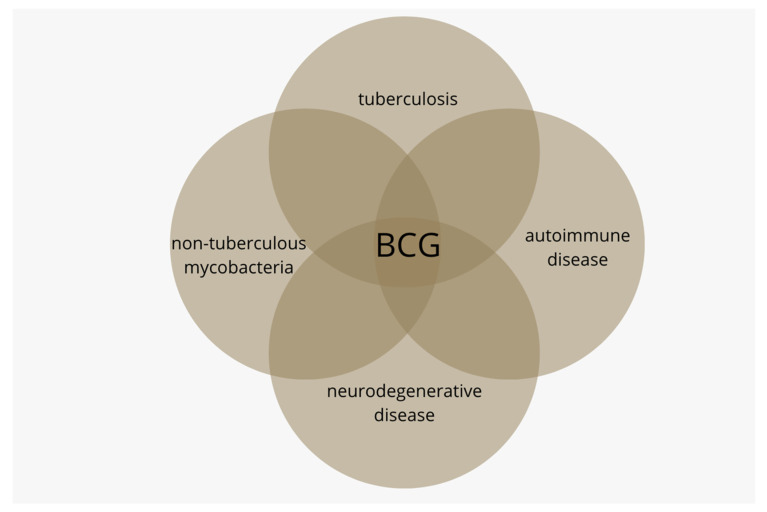
The *Bacillus Calmette-Guérin* (BCG) vaccine was originally developed to vaccinate against tuberculosis. BCG is also known to lessen the disease burden caused by non-tuberculous mycobacteria (NTM). Currently, BCG is being studied in autoimmune diseases T1D and multiple sclerosis and in neurodegenerative diseases Alzheimer’s and Parkinson’s.

**Figure 2 microorganisms-10-01919-f002:**
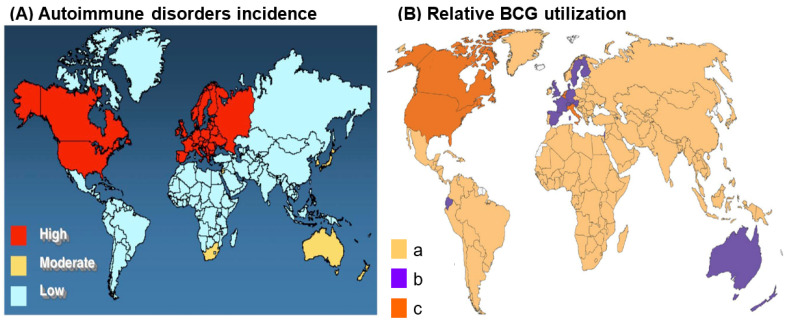
World maps displaying the relative incidence of autoimmune disorders and relative BCG utilization. (**A**) World map displaying the relative incidence of autoimmune disease in 2017. Note that the incidence is greatest in the U.S., Canada, and western Europe, followed by Australia and South Africa (https://forums.phoenixrising.me/threads/autoimmune-disease-prevalence-in-the-western-world.51642/ Accessed on 22 September 2022). Permission granted by original author, Joel Weinstock–Tufts Medical Center. (**B**) World map displaying the utilization of BCG. a: Countries with current universal BCG vaccination program. b: Countries that used to recommend universal BCG vaccination but no longer. c: Countries that never had universal BCG vaccination programs. Note that BCG utilization is least in U.S., Canada, Europe, Russia, and Australia. Permission granted by original authors (Dr. Marcel Behr-McGill University, Montreal, QC, Canada) [59]. Composite map used by permission–Dr. Dow [60].

## Data Availability

Not applicable.

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
