# Peer review of "BCG Vaccine—The Road Not Taken"

_microorganisms, 2022, doi:10.3390/microorganisms10101919_

Round 1

Reviewer 1 Report

This review describes the additional benefit of BCG vaccination which has been used to prevent miliary TB and TB meningitis in children. They pointed out that BCG vaccination provides protection against NTM infection, and BCG has some therapeutic efficacy against autoimmune diseases (T1D and MS) and neurodegenerative diseases (AD and PD). I think that this review provides some valuable information about other efficacy of BCG vaccination. But I think there is no need to the Discussion Section in the review. Instead, just include a conclusion or concluding remarks. Therefore, the contents of the Discussion should be moved into a related section. There is a big concern that MAP is related between BCG vaccination and the prevention of T1D. Because I don’t think that BCG impacts MAP infection or MAP-induced human diseases. If authors want to include MAP, the subtitle ‘BCG and MAP infection’ should be put and describe the related issues. In addition, a relation between BCG and allergic diseases should be described.  

Author Response

Dear reviewer 1.  Thank you for your constructive review and suggestions.  Although we left the Discussion section in the manuscript, I removed the entire MAP portion.  We did add a markedly abbreviated MAP portion to the NTM section.  We feel the discussion section is required to lay out the plausible mechanism of action of BCG.  We addressed BCG and allergy, citing new references.

Kowalewicz-Kulbat M, Locht C. BCG for the prevention and treatment of allergic asthma. Vaccine. 2021 Dec 8;39(50):7341-7352. doi: 10.1016/j.vaccine.2021.07.092. Epub 2021 Aug 18. PMID: 34417052.

Moulson AJ, Av-Gay Y. BCG immunomodulation: From the 'hygiene hypothesis' to COVID-19. Immunobiology. 2021 Jan;226(1):152052. doi: 10.1016/j.imbio.2020.152052. Epub 2020 Dec 24. PMID: 33418320; PMCID: PMC7833102.

Reviewer 2 Report

The review by Dow and Kidess, BCG Vaccine – The Road Not Taken, is excellent with few corrections necessary for the references. However the discussion from line 252 on MAP is out of the scope of the title and goals of this review. The discussion from line 241 to 249 covers the relevant points that related to NMT and MAP. A shorter version of discussion paragraph 6.4 could be integrated after line 249 that will render this review better focused.

In the statement in the introduction, the strong point: The associated consequences may be toned-down to indicate that causality was never yet proved in most of the examples given.

The two works of Aronson et al. JAMA. 2004;291(17):2086-2091. doi:10.1001/jama.291.17.2086   must be added to render this review more comprehensive. It shows that in the USA, BCG was efficient even in the long term (BCG vaccine efficacy persisted for 50 to 60 years) and in the second paper doi: 10.1001/jamanetworkopen.2019.12014, it shows an intriguing protection against lung cancer.

Finally, M. bovis which is mainly a zoonotic disease (DOI: 10.1111/zph.12868) is not mentioned once in this paper and it could be interesting to add in paragraph 2 (line 76) an analysis of the impact of ending BCG vaccination on M. bovis TB cases in humans.

There are some misquotes of relevant papers in this article e.g. line 164, the ref 56 is not correct as well as in line 156 the ref 57.

I advise the authors to recheck their quotes of references in the whole text.

Author Response

Dear reviewer 2. 

Thank you for your constructive review and suggestions.  We removed the entire MAP portion from the discussion section and added a markedly abbreviated MAP portion to the NTM section. 

The associated consequences was toned-down to: The suggested consequences

The work of Aronson was noted for both the durable anti-tuberculosis effect of BCG and the long-term effect against lung cancer diagnosis.  To that we added the BCG/melanoma article to the manuscript.

            Aronson NE, Santosham M, Comstock GW, Howard RS, Moulton LH, Rhoades ER, Harrison LH. Long-term efficacy of BCG vaccine in American Indians and Alaska Natives: A 60-year follow-up study. JAMA. 2004 May 5;291(17):2086-91. doi: 10.1001/jama.291.17.2086. PMID: 15126436.

            Usher NT, Chang S, Howard RS, Martinez A, Harrison LH, Santosham M, Aronson NE. Association of BCG Vaccination in Childhood With Subsequent Cancer Diagnoses: A 60-Year Follow-up of a Clinical Trial. JAMA Netw Open. 2019 Sep 4;2(9):e1912014. doi: 10.1001/jamanetworkopen.2019.12014. PMID: 31553471; PMCID: PMC6763973

            Kremenovic M, Schenk M, Lee DJ. Clinical and molecular insights into BCG immunotherapy for melanoma. J Intern Med. 2020 Dec;288(6):625-640. doi: 10.1111/joim.13037. Epub 2020 Mar 4. PMID: 32128919.

bovis zoonosis was addressed.

            Taye H, Alemu K, Mihret A, Wood JLN, Shkedy Z, Berg S, Aseffa A. Global prevalence of Mycobacterium bovis infections among human tuberculosis cases: Systematic review and meta-analysis. Zoonoses Public Health. 2021 Nov;68(7):704-718. doi: 10.1111/zph.12868. Epub 2021 Jun 24. PMID: 34169644; PMCID: PMC8487997.

            Suazo FM, Escalera AM, Torres RM. A review of M. bovis BCG protection against TB in cattle and other animals species. Prev Vet Med. 2003 Apr 30;58(1-2):1-13. doi: 10.1016/s0167-5877(03)00003-5. PMID: 12628767.

            Buddle BM, Vordermeier HM, Chambers MA, de Klerk-Lorist LM. Efficacy and Safety of BCG Vaccine for Control of Tuberculosis in Domestic Livestock and Wildlife. Front Vet Sci. 2018 Oct 26;5:259. doi: 10.3389/fvets.2018.00259. PMID: 30417002; PMCID: PMC6214331.

The reference number sequences were corrected

Round 2

Reviewer 1 Report

I think this review is successfully revised. One thing for check out is that lines 68 to 69 '~~~ routine vaccination BCG vaccination.'  It seems that the first vaccination in this sentence should be deleted.